# Tensile Overload Injures Human Alveolar Epithelial Cells through YAP/F-Actin/MAPK Signaling

**DOI:** 10.3390/biomedicines11071833

**Published:** 2023-06-26

**Authors:** Shan He, Ruihan Liu, Qing Luo, Guanbin Song

**Affiliations:** Key Laboratory of Biorheological Science and Technology, Ministry of Education, College of Bioengineering, Chongqing University, Chongqing 400030, China; heshan11@cqu.edu.cn (S.H.); 202219131169@stu.cqu.edu.cn (R.L.); qing.luo@cqu.edu.cn (Q.L.)

**Keywords:** human alveolar epithelial cells, apoptosis, F-actin, ERK1/2, JNK, tensile overload

## Abstract

Background: Explosion shockwaves can generate overloaded mechanical forces and induce lung injuries. However, the mechanism of lung injuries caused by tensile overload is still unclear. Methods: Flow cytometry was used to detect the apoptosis of human alveolar epithelial cells (BEAS-2B) induced by tensile overload, and cell proliferation was detected using 5-ethynyl-2′-deoxyuridine (EdU). Immunofluorescence and Western blot analysis were used to identify the tensile overload on the actin cytoskeleton, proteins related to the mitogen-activated protein kinase (MAPK) signal pathway, and the Yes-associated protein (YAP). Results: Tensile overload reduced BEAS-2B cell proliferation and increased apoptosis. In terms of the mechanism, we found that tensile overload led to the depolymerization of the actin cytoskeleton, the activation of c-Jun N-terminal kinase (JNK) and extracellular-signal-regulated kinase 1/2 (ERK1/2), and the upregulation of YAP expression. Jasplakinolide (Jasp) treatment promoted the polymerization of the actin cytoskeleton and reduced the phosphorylation of tension-overload-activated JNK and ERK1/2 and the apoptosis of BEAS-2B cells. Moreover, the inhibition of the JNK and ERK1/2 signaling pathways, as well as the expression of YAP, also reduced apoptosis caused by tensile overload. Conclusion: Our study establishes the role of the YAP/F-actin/MAPK axis in tensile-induced BEAS-2B cell injury and proposes new strategies for the treatment and repair of future lung injuries.

## 1. Introduction

Explosions cause injuries to important human bodily organs, harming people’s lives and health [1]. The lung is the most important respiratory organ in the human body; it mainly performs gas-exchange functions and is regulated by mechanical factors [2,3].

The overload mechanical force generated by an explosion causes injuries to lung tissue. To simulate this phenomenon, we used a mechanical stretching device to apply an overloaded tensile strain force to human alveolar epithelial cells (BEAS-2B) cells. Research has shown that mechanical stretching with different frequencies and strain sizes has different effects on lung tissue. The stretching amplitude at normal physiological levels in simulated lung tissue ranges from 5% to 12%, and amplitudes over 17% induced pathological conditions and inflammatory reactions in lung tissue, which are accompanied by acute lung injury [4]. Previous studies have confirmed that pathologically high levels of mechanical stretching (20%) induce the apoptosis of pulmonary endothelial cells, while a stretching frequency of 2 Hz reduces cells’ viability and proliferation abilities and increases the cell apoptosis rate [5,6]. Therefore, in order to simulate the pathological conditions within human alveolar epithelial cells, we used the conditions of 20% strain and 2 Hz stretching to construct the BEAS-2B cell injury system.

Generally, the perception and conversion of biomechanical signals in cells are highly dependent on the cytoskeleton [7,8]. Cell migration, adhesion, differentiation, proliferation, and apoptosis are closely related to cytoskeletal reorganization [9]. Mechanical stretching under physiological conditions promotes the rearrangement of the cytoskeleton and maintains the normal processes of cell proliferation and migration, while high-strength mechanical stretching promotes the disaggregation of the cytoskeleton and induces apoptosis. Studies have shown that actin participates in chromatin remodeling and the formation of apoptotic bodies, disrupting the formation of actin polymers in the cytoskeleton and leading to osteoblast apoptosis [10,11]. However, the relationship between the reorganization of the actin cytoskeleton and BEAS-2B cell injury under tensile overload has yet to be confirmed.

Mitogen-activated protein kinases (MAPK) are a family of serine/threonine kinases [12]. Studies have shown that the MAPK signaling pathway is related to mechanical transduction and cell damage repair [13,14]. Research on rat arterial tissue has confirmed that the response of blood vessels to stretch injuries is related to the activation of the MAPK pathway [15]. In tendon cells, the MAPK signaling pathway also induces cell migration in response to mechanical signals [16]. Therefore, the question of how the MAPK signaling pathway responds to the mechanical stimulation of overload and affects cell injury requires further study.

The Yes-associated protein (YAP) is a kind of mechanical sensor, which is regulated by mechanical signals such as the extracellular matrix, matrix hardness, tension, strain, and shear stress. These mechanical signals lead to structural changes in cells such as the cytoskeleton and nucleus, and affect cell physiological functions [17,18]. Research has shown that there is an important relationship between the expression and activity of YAP and lung injury. For example, YAP is significantly activated in sepsis-induced acute lung injury, and the inhibition of YAP expression alleviates lung injury [19]. Therefore, the matters of how YAP participates in the process of mechanical-stimulation-induced lung injury and its relationship with the cytoskeleton and MAPK signaling pathway require further attention.

In summary, our research established a system of BEAS-2B cell injury caused by tensile overload. The potential mechanism of BEAS-2B cell injury was studied, and it was found to be related to the actin cytoskeleton and the MAPK signaling pathway. These results provide new insights into the mechanism of lung injury caused by overloaded mechanical stimulation.

## 2. Materials and Methods

### 2.1. Cell Culture

The human alveolar epithelial cell line BEAS-2B was purchased from the American Type Culture Collection (ATCC). The BEAS-2B cell line was grown in 37 °C high-sugar Dulbecco modified Eagle medium (DMEM, Biological Industries, Kibbutz Beit Haeek, Israel), which was supplemented with 10% fetal bovine serum (FBS, Biological Industries), 100 μg/mL streptomycin, and 100 U/mL penicillin (Hyclone, Logan, UT, USA). The humid atmosphere contained 5% CO_2_.

### 2.2. Uniaxial Cyclic Stretch

BEAS-2B cells were seeded at a density of 2 × 10^4^ cells/cm^2^ in a chamber precoated with 5 μg/mL rat tail type I collagen (Hangzhou Shengyou Biotechnology Co., Ltd., Hangzhou, China); they were cultured for about 1 day and installed on the cell tensile loading device (Chongqing Deyansheng Technology Co., Ltd., Chongqing, China) (Figure 1A). The desired stretching frequency was selected on the human–computer interaction display screen, and the rotation of the servo motor was controlled by the controller and power box to achieve stretching changes (Figure 1B). In this study, we exposed the cells in a silicone resin chamber to a tensile treatment at 2 Hz and 20% strain. For the stretching experiments with the actin stabilizer Jasplakinolide (Jasp, 20 nM, Abcam, ab141409), the c-Jun N-terminal kinase (JNK) inhibitor SP600125 (20 μM, Sellcek, S1460), and the extracellular signal-regulated kinase 1/2 (ERK1/2) inhibitor U0126 (10 μM, HY-12031, MedChemExpress), stretching was induced for 12 h in the medium with reagents added. As a control, static cells were cultured in a small chamber under the same conditions without any stretching.

### 2.3. Protein Sample Processing and Western Blot Analysis

The total protein of the BEAS-2B cells was lysed on ice with RIPA buffer (Supplement 1× PMSF and 2% phosphatase inhibitor) (Beyotime, Shanghai, China). The protein concentration of the lysate was determined using a BCA protein determination kit (Beyotime, Shanghai, China). The dissolved substance was mixed with 5×. The loaded buffer solution was mixed and boiled at 100 °C for 10 min for denaturation. Then, 30 µg of each protein sample was separated with 10% SDS-PAGE gel and electro-sorbed to a 0.45 μm PVDF membrane (Millipore, Billerica, MA, USA). Subsequently, 5% skimmed milk was used to seal the imprint for 1h and incubated with the primary antibody at 4 °C overnight. The primary antibodies used include YAP (14074T, Cell Signaling Technology, Danvers, MA, USA), p-YAP (ab76252, Abcam, Cambridge, UK), JNK (ab179461, Abcam, Cambridge, UK), p-JNK (R26311, ZENBIO, China), ERK1/2 (ab184699, Abcam, Cambridge, UK), p-ERK1/2 (4370T, Cell Signaling Technology, Danvers, MA, USA), and GAPDH (bsm-33033M, Bios, China); they were prepared in the primary antibody diluent (Beyotime, Shanghai, China) at 4 °C overnight. After that, tris-buffered saline (TBS; Biosharp, Beijing, China) with 0.05% Tween 20 (TBST) was washed four times for five minutes each time. Then, it was incubated with the appropriate secondary antibody (ZENBIO, Chengdu, China) and 5% skim milk at room temperature for 1 h. Finally, the membrane was washed four times with TBST and the protein expression was assessed every five minutes using an ECL blot analysis system (Bio-OI, Guangzhou, China).

### 2.4. Cell Proliferation Assay

The 5-ethynyl-2-deoxyuridine (EdU) incorporation was used to detect the proliferation of BEAS-2B cells. EdU was added to the cultured cells and cultured for another 2 h. The EdU-488 cell proliferation test kit (Beyotime, Shanghai, China) was used to evaluate the proliferation according to the manufacturer’s instructions.

### 2.5. Cell Viability Assay

The CCK8 (Biosharp, Beijing, China) experiment was used to detect the effect of tensile overload on the viability of the BEAS-2B cells. After the tensile overload treatment, each chamber was added with a uniformly mixed CCK8 solution and culture medium in a ratio of 1:10. After incubation in an incubator for 2 h, the remaining solution was transferred to a 96-well plate, and the absorbance at 450 nm was measured using an enzyme scale.

### 2.6. Apoptosis Assay

Apoptosis was analyzed using an Annexin V-FITC/PI analysis kit (Beyotime, Shanghai, China). The cells were collected and washed twice with PBS; then, the binding solution was added according to the instructions, and 5 μL Annexin V-FITC solution with 10 μL propidium iodide (PI) was incubated in the dark for 10 min. The samples were then analyzed by flow cytometry. The percentage of apoptotic cells is the sum of the percentage of early and late apoptotic cells [20].

### 2.7. Immunofluorescence Assay

The BEAS-2B cells were immersed in 4% paraformaldehyde solution and fixed for 15 min, washed with PBS 3 times for 5 min each time, infiltrated with 0.3% Triton X-100 (Solarbio, Beijing, China)/PBS for 10 min, then washed with PBS 3 times for 5 min and sealed with 1% BSA (Solarbio, Beijing, China)/PBS for 1 h. The actin filament was stained with Phalloidin (Solarbio, Beijing, China) and DAPI (Biosharp, Beijing, China). The images were taken with a Leica fluorescence microscope (Leica, Solms, Germany).

### 2.8. Cell Transfection

We designed and synthesized siRNA for YAP and the control (Tsingke, Beijing, China), which was transfected into the BEAS-2B cells using the Lipofectamine 3000 reagent (Invitrogen, Carlsbad, CA, USA) according to the manufacturer’s instructions. The siRNA sequences used in this study are as follows: siYAP-1: GACAUCUUCUGGUCAGAGA UCUCUGACCAGAAGAUGUC; siYAP-2: CUGGUCAGAGAUACUUCUU AAGAAGUAUCUCUGACCAG.

### 2.9. Bioinformatics Analysis

To investigate the molecular mechanism of the effect of stretching on BEAS-2B cells, we retrieved the GSE16650 dataset from the GEO database, used GEO2R for the differential analysis, and used KEGG to enrich the related pathways.

### 2.10. Statistical Analysis

The data were analyzed using GraphPad Prism version 8 (GraphPad Software). For all analyses, the student’s *t*-test was used for comparisons between the two groups. *p* < 0.05 was calculated as statistically significant.

## 3. Results

### 3.1. Tensile Overload Induces BEAS-2B Cell Apoptosis and Inhibits Cell Proliferation

To study the effect of tensile overload on BEAS-2B cells, we used a strain amplitude and frequency of 20% strain at 2 Hz for cell stretching. The results showed that, when exposed to 20% of the strain and stretched at 2 Hz for 6 h and 12 h, the stretched BEAS-2B cells showed increased apoptosis (Figure 2A,B) and decreased cell viability (Figure 2C) and cell proliferation (Figure 2D,E) compared to the non-stretched group.

### 3.2. Tensile Overload Inhibits the Polymerization of Cytoskeletal F-Actin and Induces BEAS-2B Apoptosis

The actin cytoskeleton is closely related to the transmission of intracellular mechanical signals. To explore the role of the actin cytoskeleton in BEAS-2B cell injury induced by tensile overload. We evaluated the response of the F-actin cytoskeleton to tensile overload. The results showed that, compared to the non-stretching group, a tensile overload of 20% strain and 2 Hz could result in a thinning of the actin stress fibers as the stretching time increased (Figure 3A). The same results were obtained in the Western blot experiment (Figure 3B). After the addition of the actin polymerization drug Jasp (20 nM), the actin stress fibers were effectively stabilized (Figure 3C). Flow cytometry showed that, in the presence of Jasp, apoptosis in cells in the stretching group was significantly reduced (Figure 3D,E). These findings suggest that tensile overload regulates BEAS-2B cell injury through the depolymerization of actin stress fibers. 

### 3.3. Tensile Overload Reduces Actin Stress Fibers and Activates JNK and ERK1/2 Signaling

In order to determine the regulatory mechanism of tensile overload on BEAS-2B cell damage, we analyzed the transcriptome results of GSE16650 using bioinformatics methods. The results of the KEGG analysis indicate that the MAPK signaling pathway plays a major role (Figure 4A). ERK1/2, JNK, and p38 are three subfamilies of MAPK. Subsequently, we evaluated the activity of JNK, ERK1/2, and p38 in tensile-overloaded BEAS-2B cells using Western blotting. We observed that the expression of p-JNK and p-ERK1/2 was significantly upregulated over time (Figure 4B). However, the expression of p-p38 did not change (Appendix A). Next, we attempted to determine the relationship between the F-actin cytoskeleton and p-JNK and p-ERK1/2. Therefore, we added Jasp to detect the expression levels of p-JNK and p-ERK1/2 after stretching. The results showed that, after adding Jasp, the expression of p-JNK and p-ERK1/2 decreased (Figure 4C). In addition, we evaluated the effects of the JNK inhibitor SP600125 (20 μM) and the ERK1/2 inhibitor U0126 (10 μM) on BEAS-2B cell injury. We noted that the addition of SP600125 and U0126 inhibited the phosphorylation levels of JNK and ERK1/2 (Figure 4D,E). Consistently with this finding, flow cytometry showed that inhibiting the phosphorylation of JNK and ERK1/2 significantly reduced the rate of cell apoptosis (Figure 4F,G). These results demonstrate that JNK and ERK1/2 mediation of the F-actin cytoskeleton plays an important role in the injury of BEAS-2B cells induced by tensile overload.

### 3.4. YAP Downregulation Induces the Polymerization of F-Actin in the Cytoskeleton and Inhibits the Activation of JNK and ERK1/2

YAP is closely related to changes in the actin cytoskeleton and the MAPK signaling pathways. To determine the changes in YAP in BEAS-2B injuries induced by tensile overload, we applied a tensile force of 20% strain and 2 Hz to BEAS-2B cells for 6 h and 12 h. The results showed that the expression of YAP increased with the increasing stretching time (Figure 5A). We then evaluated the level of YAP phosphorylation because it is related to the intracellular localization of YAP. The results of the Western blotting showed that YAP phosphorylation was downregulated (Figure 5B). These results indicate that tensile overload induces the expression of YAP in the nucleus of BEAS-2B cells.

To further investigate whether YAP can directly regulate the proliferation and apoptosis of BEAS-2B cells, we evaluated the biological significance of YAP expression in BEAS-2B cell lines by silencing the expression of YAP. siRNA knockout was used to reduce the expression of the YAP protein 48 h after YAP transfection (Figure 5C). We further examined the changes in the F-actin, JNK, and ERK1/2 activity of the cytoskeleton after inhibiting the expression of YAP. The results showed that F-actin polymerized after YAP inhibition (Figure 5D). Subsequently, JNK and ERK1/2 activities were inhibited (Figure 5E). In YAP knockout cells exposed to tensile overload, apoptosis significantly decreased (Figure 5F,G). The results suggest that tensile overload regulates the apoptosis of BEAS-2B cells through the YAP/F-actin/MAPK signaling axis.

## 4. Discussion

A major cause of death is the expansion-related injury of lung tissue caused by explosions [21]. However, few studies have reported the regulatory mechanism of expansion injuries in lung tissue, and little is known at the cellular and molecular levels [22,23]. Therefore, it is necessary to simulate overload-induced tensile strain in vitro and investigate the biological mechanisms of lung injury.

Mechanical stretching with different frequencies and strain sizes has different effects on lung tissue [24]. Studies have shown that the stimulation of pathological mechanical stretching induces apoptosis [25]. ATII cells have shown that ROS production and cytochrome C increase in mechanically stretched cells, while the mitochondrial membrane potential decreases and promotes apoptosis [26]. ATII primary alveolar epithelial cells were isolated from C57BL/6J wild-type mice, cultivated, and subjected to 15% strain stretching for 24 h; they showed increased alveolar permeability and cell death [27]. Moreover, periodic stretching can also induce the apoptosis of human periodontal ligament cells by activating caspase-5 [28]. This is consistent with our experimental results, which indicate that overloaded mechanical strain causes significant injury to BEAS-2B cells. Applying strain of 2 Hz at 20% promotes the apoptosis of BEAS-2B cells and reduces their vitality and proliferation abilities.

The cytoskeleton, which includes microfilaments, microtubules, and stress fibers, is a structure that is important for cells to maintain cell morphology and respond to mechanical stimuli [29]. It is also an important medium for cell force signal transduction, signal material transmission, and protein transportation [5]. F-actin is an important component of cytoskeletal filaments [30]. Actin cytoskeletal networks can provide mechanical support for cells and determine cell shapes; they can drive cell movement to determine cell migration and division [31,32]. Research has found that F-actin functions on microfilaments through continuous polymerization and depolymerization, and it communicates mechanical signals inside and outside cells [33]. In addition, studies have shown that mechanical stretching under overload can induce F-actin depolymerization and affect the normal physiological functions of cells. In our study, tensile overload at 2 Hz with 20% strain decreased the expression of F-actin. The immunofluorescence results showed that the number of filamentous fiber bundles in BEAS-2B cells decreased and the skeleton structure was relatively loose. Jasp is an actin stabilizer that can be used to promote the aggregation of cytoskeletal actin and protect it from damage [34]. It was found that, after the cells were treated with Jasp, the aggregated actin fiber bundles in the cells increased [35]. After adding Jasp, the fluorescence intensity was restored, the number of filamentous fiber bundles in the cells increased, the stability increased, the level of apoptosis decreased. These results further indicate that the addition of Jasp increases the polymerization of cytoskeleton F-actin and reduces the apoptosis induced by tensile overload.

Through a bioinformatics analysis of GSE16650, we found that the MAPK signaling pathway plays a major role in it. Other studies have reported that physiological mechanical stretching promotes tendon cell migration, which involves the MAPK signaling pathway [16]. However, many studies have reported that the activation of the MAPK pathway is positively correlated with the degree of cell damage. ERK1/2 can be involved in regulating the pro-inflammatory response induced by cell stretching and is associated with cell activity [36]. In VSMC cells, overloaded mechanical stretching can activate JNK signaling and induce apoptosis [37]. In our study, tensile overloading activated JNK and ERK1/2 signaling. After adding SP600125, an inhibitor of JNK, and U0126, an inhibitor of ERK1/2, cell apoptosis decreased. This result further indicates that tensile overload can induce BEAS-2B cell injury by upregulating the phosphorylation expression of JNK and ERK1/2. Some studies have found that the MAPK signaling pathway is involved in the process of the cytoskeleton’s regulation of cellular physiological functions in the repair of lung injury [12]. Our findings are similar in that the addition of Jasp inhibited the phosphorylation expression levels of JNK and ERK1/2, which were elevated by tensile overload. This result indicates that tensile overload promotes the depolymerization of F-actin in the cytoskeleton, affects the activation of JNK and ERK1/2, and induces the apoptosis of BEAS-2B cells.

YAP is a mechanical regulatory factor that is widely involved in various mechanical regulatory pathways [38,39]. Many studies have shown that changes in the cytoskeleton are closely related to the expression of YAP, and the depolymerization of the cytoskeleton can promote the transfer of YAP to the cytoplasm and its inactivation [40]. However, some studies have shown that YAP can negatively regulate actin expression. After inhibiting YAP expression, the fluorescence intensity of F-actin increases, further affecting downstream factors [41]. We believe that the occurrene of these differences may be related to differences in cell type, cell inoculation density, and the force exerted on cells [42]. Our results indicate that tensile overload can promote the expression of YAP and induce its nuclear transfer. After using siRNA to inhibit the expression of YAP, BEAS-2B cells showed a decrease in apoptosis. Subsequently, we further examined the changes in cytoskeletal F-actin, JNK, and the ERK1/2 pathways after the inhibition of YAP expression. The results showed that YAP silencing promoted the polymerization of F-actin and inhibited the activity of JNK and ERK1/2. Moreover, we believe that, with the continuous development of organoid technology and other technologies in future research, we can build a lung organoid model to explore the damage of overload stretching on the lung tissue level. At the animal level, mechanical ventilation and other methods were used to simulate the overload stretching effect, further validating the conclusions obtained in this study in vivo. Our research also has some limitations. For example, we only studied the role of BEAS-2B cells in overload-tensile-induced injury, while other human cells and tissues require further understanding and exploration.

## 5. Conclusions

Here, we elucidate a molecular pathway whereby tensile overload affects the reduction of F-actin stress fibers and the MAPK pathway by up-regulating the expression of YAP, which may further lead to BEAS-2B cell injury. In summary, our research suggests that the YAP/F-actin/MAPK signal cascade may be the mechanism of mechanical-tension-induced BEAS-2B cell injury. This study provides new insights into mechanical-force-induced lung injury and provides new strategies for the treatment and repair of future lung injury (Figure 6).

## Figures and Tables

**Figure 1 biomedicines-11-01833-f001:**
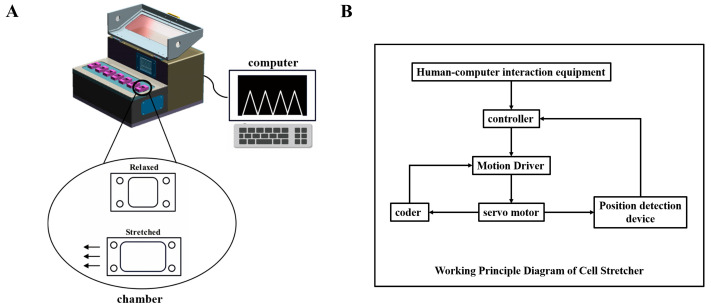
Cell tensile loading device. (**A**) Design drawing of the cell tensile loading device. (**B**) Diagram of the device’s working principles.

**Figure 2 biomedicines-11-01833-f002:**
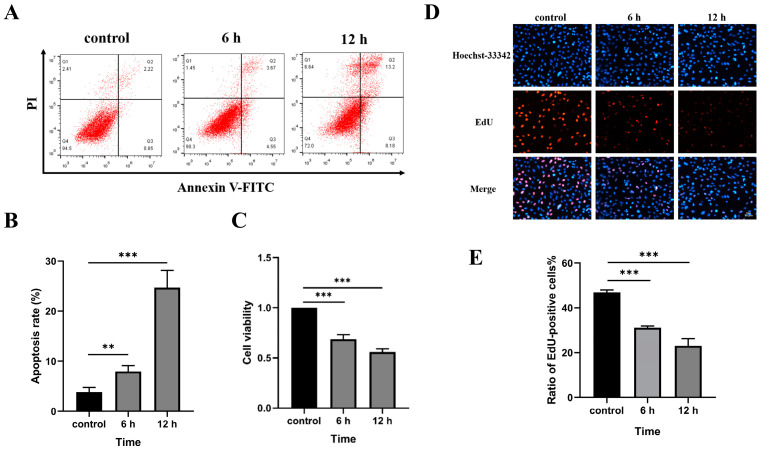
BEAS-2B cell injury induced by 20% strain and 2 Hz tensile overload. (**A**) Flow cytometry detection of tensile overload on apoptosis. (**B**) The apoptosis was counted and quantized. (**C**) CCK8 detection of tensile overload on cell viability. (**D**) EdU detection of tensile overload on cell proliferation (scale bar, 50 μm). (**E**) Counting and quantifying cell proliferation. The graph shows the mean with SD; *n* = 3, ** *p* < 0.01; *** *p* < 0.001.

**Figure 3 biomedicines-11-01833-f003:**
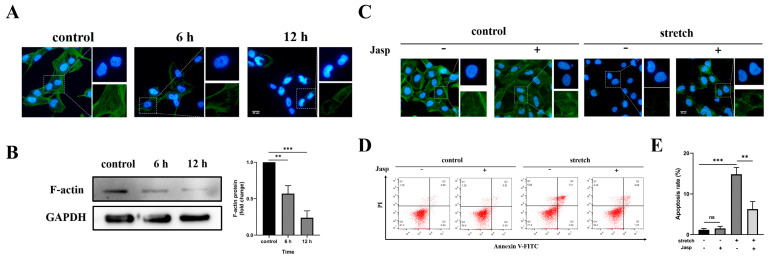
Tensile overload induces BEAS-2B cell injury through cytoskeleton polymerization. (**A**) The F-actin organization was detected via phalloidin staining; nuclei were stained with DAPI (blue) (scale bar, 25 μm). (**B**) Western blotting was used to analyze the expression of F-actin. (**C**) Jasp combined with tensile overload for 12 h on cytoskeletal F-actin was detected by an immunofluorescence assay (scale bar, 25 μm). (**D**) Flow cytometry was used to detect the Jasp combined with tensile overload on cell apoptosis for 12 h. (**E**) The apoptosis was counted and quantized. The graph shows the mean with SD; *n* = 3, ** *p* < 0.01; *** *p* < 0.001.

**Figure 4 biomedicines-11-01833-f004:**
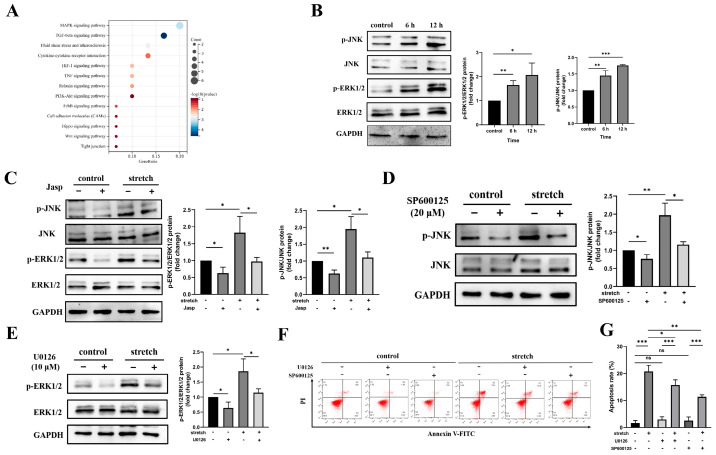
JNK and ERK1/2 mediated stretching on the F-actin cytoskeleton and BEAS-2B cell apoptosis. (**A**) KEGG analysis of stretch-related pathways in BEAS-2B cells. (**B**) Western blot detection of p-JNK, JNK, p-ERK1/2, and ERK1/2 expression and statistical results. (**C**) Western blot detection and statistical results for the expression of p-JNK, JNK, p-ERK1/2, and ERK1/2 with tensile overload and Jasp combined for 12 h. (**D**) Western blotting was used to detect the expression and statistical results of p-JNK and JNK when overstretching was co-treated with SP600125. (**E**) Western blotting was used to detect the expression and statistical results of p-ERK1/2 and ERK1/2 when overstretching was co-treated with U0126. (**F**) U0126 and SP600125 were co-treatment with tensile overload on the apoptosis of BEAS-2B cells by flow cytometry for 12 h. (**G**) The apoptosis was counted and quantized. The graph shows the mean with SD; *n* = 3, * *p* < 0.05; ** *p* < 0.01; *** *p* < 0.001, ns: non-significance.

**Figure 5 biomedicines-11-01833-f005:**
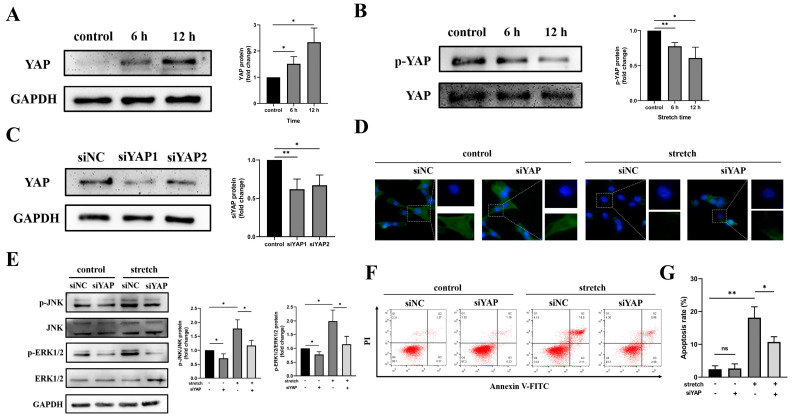
YAP downregulation reduces BEAS-2B cell injury caused by tensile overload. (**A**) Western blot detection of tensile overload on YAP content and statistical results. (**B**) Western blot detection of tensile overload on p-YAP content and statistical results. (**C**) Western blot detection of siRNA (siYAP1, siYAP2) knockout YAP expression and statistical results in BEAS-2B cells. (**D**) Immunofluorescence assay for F-actin remodeling evaluation after tensile overload treatment in BEAS-2B cells with siNC and siYAP (scale bar, 25 μm). (**E**) Western blotting detection and statistical analysis of p-ERK1/2, ERK1/2, p-JNK, and JNK protein expression in BEAS-2B cells transfected with siNC and siYAP. (**F**) Detection of changes in apoptosis after the inhibition of YAP by flow cytometry. (**G**) The apoptosis was counted and quantized. The graph shows the mean with SD; *n* = 3, * *p* < 0.05; ** *p* < 0.01, ns: non-significance.

**Figure 6 biomedicines-11-01833-f006:**
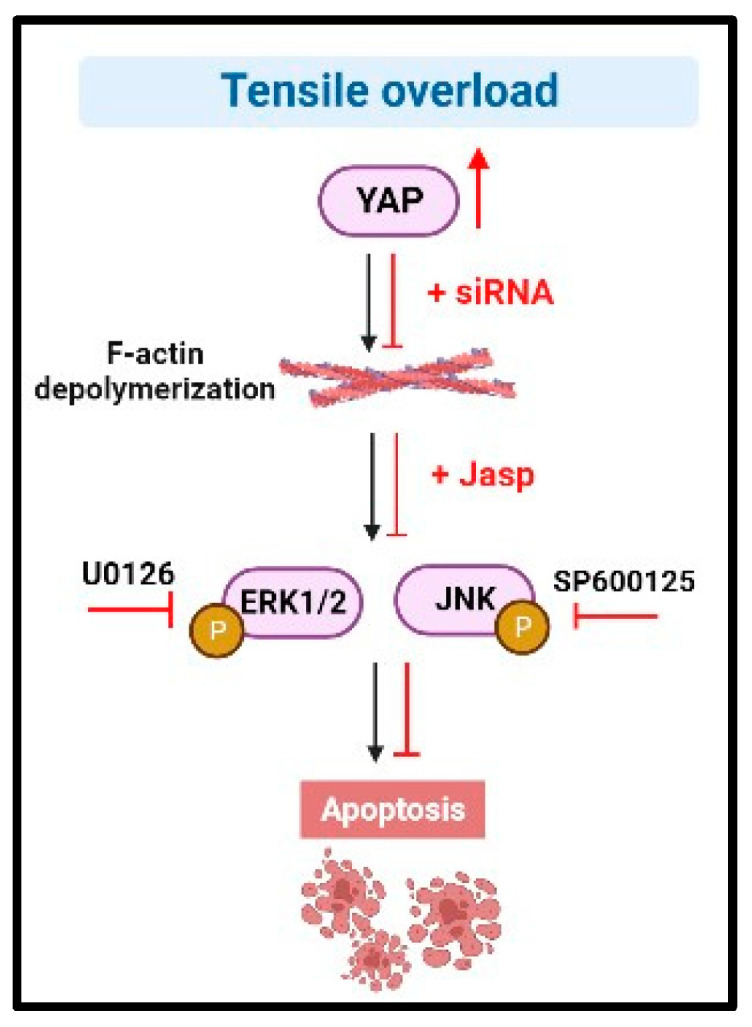
A proposed schematic diagram of BEAS-2B cell injury induced by tensile overload. We found that 2 Hz mechanical stretching with 20% strain can induce apoptosis and inhibit proliferation. Tensile-overload-induced F-actin depolymerization upregulated the expression of p-ERK1/2 and p-JNK, which in turn induced apoptosis in BEAS-2B cells. Jasp treatment could promote F-actin polymerization, inhibit the expression of p-ERK1/2 and p-JNK, and reduce the apoptosis of BEAS-2B cells. In addition, tensile overload promoted the expression and activity of YAP, F-actin was polymerized after silencing YAP, and the expression of p-ERK1/2 and p-JNK was inhibited. These results confirm that tensile overload can induce the apoptosis of BEAS-2B cells through the YAP/F-actin/MAPK signaling axis.

## Data Availability

All data are contained within the article.

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
