# Peer review of "Tensile Overload Injures Human Alveolar Epithelial Cells through YAP/F-Actin/MAPK Signaling"

_biomedicines, 2023, doi:10.3390/biomedicines11071833_

Round 1

Reviewer 1 Report

In this study, the authors have studied the role of the YAP/F-actin/MAPK axis in tensile-induced BEAS-2B cell injury by analyzing the f-actin protein by immune blotting, flow cytometry, etc.  They also used Jasp to confirm the role of f-actin in tensile-induced epithelial cell injury. Though this study is interesting, it has some limitations and requires more data to support the conclusions drawn. The English language and grammatical/syntax errors need to be fixed, and it is very hard to read and understand in some areas.

Some general and specific comments

Provide a proper source to purchase BEAS-2B cells.  Typical Culture Preservation Center of the United States seems inappropriate.

Write the following methods in past tense: Uniaxial cyclic stretch (Select the desired, Cultivate static cells etc)

Provide the source, model name, and place of the cell tensile loading device.

Always use the capital W for "western blot.

All the methods need to be rewritten in past tense and English, and grammatical errors need to be fixed.

synthesised by tsingke-provide full information about tsingke?

How do you translate this to animal models, and then to clinical settings. Though the in vitro study is interesting, it is important to discuss its role in translating these results to the next stage.

In discussion: Our results are similar. repeated a couple of times. Rewrite this sentence and combine it with the next sentence, as this one is a very small one.

As per the KEGG analysis, MAPK is the major pathway, and next is TGF-b. It is suggested to study at least one or two TGF-b signaling pathways in this in vitro model, as this study focused on only one aspect and used only one cell line. It is also suggested to have some data on primary Type 2 alveolar epithelial cells isolated from the mice, and performing similar experiments would add more evidence and support the mechanism studied in the present study

Extensive language editing is requires. Methods section and result, discussion sections are fragmented and need to be fixed.

Reviewer 2 Report

       The current manuscript concerns with the role of the YAP/F-actin/MAPK axis in tensile-induced BEAS-2B cell injury and proposes new strategies for the treatment and repair of future lung injury. It is well-planned study, and the work is interesting and can be considered for publication after minor revision.

Suggesting the authors to address these issues:

1-    Avoid the abbreviations in the abstract section.

2-    The authors should add the sources and purity percentage for all chemical compounds used in the current study.

3-     In page 4, (2.6. Apoptosis assay), the authors should support the assay by adding the following reference https://doi.org/10.3390/molecules28104220.

4-    Please adjust the quality and resolution of all figures.

5-    Summarizing all findings of the current study in the conclusion section is highly recommended.

6-    Graphical abstract for the present study is recommended.

7-    Please adjust all the references according to the journal’s instructions.

Overall, the manuscript can be considered for publication after minor revision as indicated above.

Reviewer 3 Report

Dear the Editor

He S et al reported the effect of tension on YAP/F-actin/MAPK signaling pathway. In this study, a novel device that transduces tension to the culture cells was developed (Fig. 1). Using this original apparatus, these authors reported that overload of tension on Beas-2B cells induced enhanced apoptosis (Fig. 2). Under this experimental condition, phosphorylation of JNK and Erk was elevated (Fig. 4). Finally, these authors found that down-regulation of YAP induced activation of JNK and Erk, respectively, demonstrating that YAP/F-actin/MAPK signaling pathway is invovled in the epithelial injury by tension. Overall, this manuscript concisely described experimentl output. Typos need to be extensively corrected throughout manuscript.

Major concerns:

1) Please provide the reason why this tension condition (ie 20% strain and 2Hz cell stretching) has been selected. In other words, is this or similar condition occasionally found under physiological conditions?

Minor concerns:

1) Because there was no data for p38 in all Figures in the main text, description for p-p38 and p38 seemed to be better eliminated (LL104-105).

2) Manufacture of CCK8 (L118), siRNA (LL143-144), and GraphPad Prism (L-150) is missing. Please look manuscript carefully and also provide all details for regaents if there is missing other than the above in Materials and Method section.

3) In L200 and 215, sp should be capitalized.

4) Pleaase explain Jasp (L174). This is not described in M&M section.

Round 2

Reviewer 1 Report

Authors have adequately answered my comments. No further comments..